# Model-Based Control of a 4-DOF Rehabilitation Parallel Robot with Online Identification of the Gravitational Term

**DOI:** 10.3390/s23052790

**Published:** 2023-03-03

**Authors:** Rafael J. Escarabajal, José L. Pulloquinga, Vicente Mata, Ángel Valera, Miguel Díaz-Rodríguez

**Affiliations:** 1Departamento de Ingeniería de Sistemas y Automática, Instituto de Automática e Informática Industrial, Camino de Vera s/n, 46022 Valencia, Spain; 2Centro de Investigación en Ingeniería Mecánica, Departamento de Ingeniería Mecánica y de Materiales, Universitat Politècnica de València, Camino de Vera s/n, 46022 Valencia, Spain; 3Departamento de Tecnología y Diseño, Facultad de Ingeniería, Núcleo la Hechicera, Universidad de los Andes, Merida 5101, Venezuela

**Keywords:** adaptive control, dynamic parameter identification, relevant parameters, parallel robot

## Abstract

Parallel robots are being increasingly used as a fundamental component of lower-limb rehabilitation systems. During rehabilitation therapies, the parallel robot must interact with the patient, which raises several challenges to the control system: (1) The weight supported by the robot can vary from patient to patient, and even for the same patient, making standard model-based controllers unsuitable for those tasks since they rely on constant dynamic models and parameters. (2) The identification techniques usually consider the estimation of all dynamic parameters, bringing about challenges concerning robustness and complexity. This paper proposes the design and experimental validation of a model-based controller comprising a proportional-derivative controller with gravity compensation applied to a 4-DOF parallel robot for knee rehabilitation, where the gravitational forces are expressed in terms of relevant dynamic parameters. The identification of such parameters is possible by means of least squares methods. The proposed controller has been experimentally validated, holding the error stable following significant payload changes in terms of the weight of the patient’s leg. This novel controller allows us to perform both identification and control simultaneously and is easy to tune. Moreover, its parameters have an intuitive interpretation, contrary to a conventional adaptive controller. The performance of a conventional adaptive controller and the proposed one are compared experimentally.

## 1. Introduction

Parallel robots (PRs) consist of closed kinematic chains involving a fixed and a mobile platform, where an end-effector is defined and controlled. They have extensive applications [1] as they present several advantages over their serial counterparts regarding accuracy and dynamic behavior with medical applications being one of the most notable [2,3]. Conventional lower limb rehabilitation is complicated and requires intensive work; thus, robotic assistance is increasingly becoming a solution to complement conventional protocols [4]. Regarding lower limb rehabilitation tasks, devices such as gait trainers [5] and ankle rehabilitation robots [6] have been widely studied. In this research, we consider a PR with four degrees of freedom (DOF) developed for knee rehabilitation and diagnosis purposes.

A mechanical design of a human knee reeducation mechanism was presented in [7]. In [8], a control system implemented on a PR was designed for lower limb rehabilitation of bedridden stroke survivors. Parallel robotic manipulators can be kinematically optimized by design to avoid singularities [9], to achieve accuracy even with potential changes in the geometric parameters during performance [10], or with task-specific criteria using a performance index [11].

Robots can be controlled in various ways. In particular, PID-based controllers predominate in industrial environments [12,13]. These controllers do not generally use a dynamic model. In manipulations involving interactions with a human similar to those considered in this study, a model-based control technique is better suited [14]. For example, Computed-Torque Control [15,16] is an approach that uses the inverse dynamic model to compensate for all the dynamic effects in real time. However, this kind of scheme requires precise model identification [17]. An important limitation of standard model-based control is that it cannot deal with parameter changes, in that if any of the inertial (including the payload) or friction parameters change online, the model remains the same, becoming inaccurate. The variation in parameters may be due to a natural degradation of the mechanism, unmodeled or non-accurate operational regime, and manual change of the payload. Rehabilitation therapies with human–robot interaction are affected by this issue since the force exerted on the robot can vary among patients and even for the same patient, so standard model-based controllers become inaccurate.

It is well-known that the rigid body dynamic model is linear in the inertial parameters [18], and the problem of time-dependent dynamic parameters has been tackled using robust control [19] and adaptive control [20], which are approaches commonly used where the controller tries to respond to uncertainties. However, it may be that not all the dynamic parameters estimated by the adaptive controller are useful as only a subset of those parameters contributes to the behavior of the robot. The subset of reduced dynamic parameters is known as base parameters. Gautier [21] proposed Singular Value Decomposition (SVD) to identify the base parameters of a PR. However, a PR struggles to excite useful trajectories properly due to the constrained range of movements that characterize these robots [22,23]. Díaz-Rodríguez et al. [24] reduced the set of base parameters to a new subset of relevant parameters considering physical feasibility. The relevant parameters lead to simpler, more robust models and fewer computational requirements for an adaptive controller, while keeping the linear relationship of the model with respect to the relevant parameters.

Recent research has been conducted regarding the identification of relevant parameters [24] and adaptive control with real-time updates [25] applied to a 3-DOF parallel manipulator for rehabilitation purposes with a variable payload. Duan et al. [26] proposed a method to identify a payload on a serial robot based on trajectory excitation to compensate for gravity and inertial force. However, they relied on the use of an external force sensor to perform the identification. Kim et al. [27] implemented a system identification method for a delta robot to estimate a set of uncertain parameters to be included in the dynamic model, although it did not experience significant payload changes.

Motivated by the aforementioned studies, the contribution of this paper is the development of a model-based controller comprising a proportional-derivative (PD) controller and an online relevant parameter identifier for the gravitational model applied to a 4-DOF knee rehabilitation PR. These relevant parameters are extracted to improve the condition of the linear model such that simple linear regression techniques can be robustly applied online to get an accurate measure of the gravitational term. Firstly, an offline identification of the relevant parameters is carried out by applying the SVD process to a set of knee rehabilitation trajectories. The second step involves the selection of the relevant parameters considering physical feasibility. The subsequent online estimation of these relevant parameters is performed by using the reduced model obtained from the SVD process and a least squares algorithm that exploits the well-conditioned problem. In particular, the moving-window least squares and recursive least squares methods are implemented to create two model-based controllers that are successfully applied to both simulations and the actual PR.

The experiments performed are designed specifically for knee rehabilitation purposes. In control tasks, this identification allows to keep the error stable after a payload change, while at the same time estimate such payload without requiring a force sensor since it is shown to be included within the relevant parameters, which would turn very convenient for estimating the patient leg’s weight in the rehabilitation context. Moreover, the parameters of the regression are few, as well as intuitive and easy to tune, in contrast to the more involved tuning required for a conventional adaptive controller. The results of the proposed controllers are evaluated along with an adaptive controller designed for the same PR to show the main advantages of the new proposed algorithms.

This paper is organized as follows: Section 2 introduces the 4-DOF PR, its dynamic model, the base parameters identification process based on the SVD procedure, and the relevant parameter selection. Section 3 summarizes the adaptive controller for the PR under study and presents the two variants of the proposed controller. Section 4 describes the experimental procedure and results. Finally, the main conclusions are summarized in Section 5.

## 2. Parallel Robot for Knee Rehabilitation and Diagnosis

### 2.1. Knee Rehabilitation and Diagnosis

The purpose of the PR is the rehabilitation of knee injuries but also the diagnosis of Anterior Cruciate Ligament (ACL)-deficient knee. Additionally, it could be applied to the rehabilitation of ankle injuries.

In clinical rehabilitation, knee rehabilitation tasks involve flexion–extension movements of the leg in the tibiofemoral plane. In some cases, the flexion–extension of the knee is combined with hip flexion [28]. During the knee rehabilitation procedure, the ankle is adapted to the best physiological pose of the patient, so its movement must be controlled.

The Anterior Cruciate Ligament (ACL) limits the translational and rotational movements of the tibia with respect to the femur. Consequently, knee instability is an accepted indicator of ACL damage. Specific tests have been developed to detect it. The two most recognized are:The Lachman test: Relative displacements of the tibia/femur in the tibiofemoral plane [29].The Pivot Shift Test: Reproduce translational and rotational instability in the knee in the coronal plane [30].

Based on the fundamental movements for clinical rehabilitation and diagnosis of the knee, the movements required for robotic devices involved in this task are (Figure 1):M1Flexion of the leg in the direction perpendicular to the tibiofemoral plane.M2Rotation of the leg along an axis perpendicular to the PR mobile platform.M3Translation in the direction of the X-axis, contained in the tibiofemoral plane.M4Translation in the direction of the Z-axis, contained in the tibiofemoral plane.

**Figure 1 sensors-23-02790-f001:**
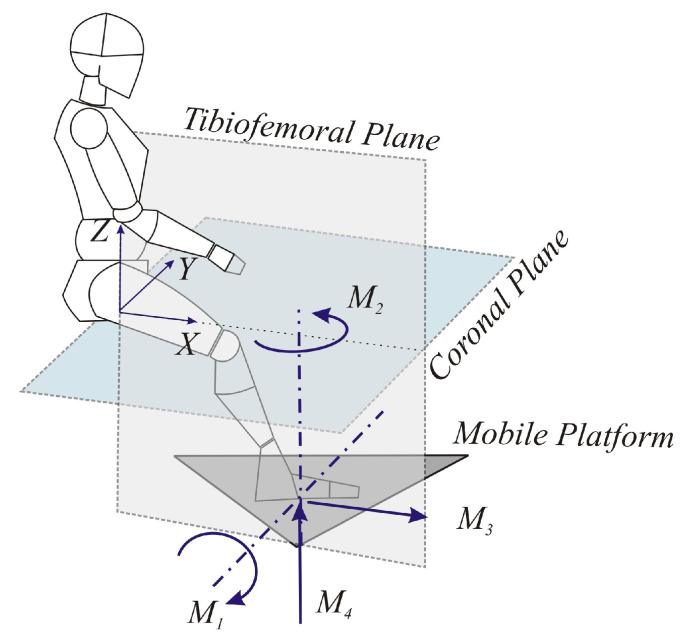
Movements required for knee rehabilitation.

The movements M3 and M4, which must be disengaged, allow the development of knee rehabilitation tasks. For this purpose, M1 is essential to achieve a proper position of the leg during rehabilitation/diagnosis tasks. In addition, this movement could perform flexion and/or extension of the ankle in the tibiofemoral plane. The Lachman test can be done primarily by the M3 movement. The Pivot Shift Test is specifically performed by the M2 movement. The previous movements can be achieved with the 4-DOF PR designed and built at the Universitat Politècnica de València [31].

### 2.2. Architecture

The 4-DOF PR to perform rehabilitation and diagnosis is shown in Figure 2. It has a 3UPS+RPU architecture, meaning that the mobile platform is connected to the fixed platform using three external limbs in UPS configuration and a central limb in RPU configuration. The letters R, P, U and S stand for the revolute, prismatic, universal, and spherical joints, respectively. In addition, the actuated joints are underlined.

The position of each joint is defined by 11 generalized coordinates ($q_{11}$, $q_{12}$, $q_{13}$, …, $q_{41}$, $q_{42}$), where the first subindex corresponds to the limb and the second refers to the joint. The 4 DOF of the mobile platform are expressed by four generalized coordinates ($x_{m}$, $z_{m}$, θ, ψ), see Figure 2. The entire set of generalized coordinates are classified as independent, i.e., actuated. In particular, they correspond to the prismatic joints (${\vec{q}}_{ind}$). The secondary coordinates are the passive joints and the mobile platform coordinates (${\vec{q}}_{s}$). The set of coordinates is defined as follows:(1)$$\begin{array}{cc} {\vec{q}}_{s} & ={\left[q_{11}\;q_{12}\;q_{21}\;q_{22}\;q_{31}\;q_{32}\;q_{41}\;x_{m}\;z_{m}\;\theta \;\psi \right]}^{T}\\ {\vec{q}}_{ind} & ={\left[q_{13}\;q_{23}\;q_{33}\;q_{42}\right]}^{T} \end{array}$$

In this study, the 15 generalized coordinates (N=15) are organized as follows:(2)$$\vec{q}={\left[{\vec{q}}_{s}\;{\vec{q}}_{ind}\right]}^{T}$$

Both the inverse and forward kinematic analysis of the considered robot have been addressed in [32].

The fundamental movements designed for rehabilitation based on the 3UPS+RPU PR are depicted in Table 1.

### 2.3. Dynamic Model

The dynamic model of the 3UPS+RPU PR is determined by the Principle of Virtual Power and applying D’Alembert’s Principle. Considering that the PR is modelled by a set of generalized coordinates, the equation of motion of the 3UPS+RPU PR is given by:(3)$$M_{N\times N}\cdot {\vec{\ddot{q}}}_{N\times 1}+Q_{cyc_{N\times N}}\cdot {\vec{\dot{q}}}_{N\times 1}+{\vec{Q}}_{grav_{N\times 1}}-{\vec{Q}}_{act_{N\times 1}}-J_{q_{11\times N}}^{T}\cdot {\vec{\lambda }}_{11}=\vec{0}$$
where:
$\vec{\dot{q}}$:Vector of generalized velocities.$\vec{\ddot{q}}$:Vector of generalized accelerations.*M*:PR mass matrix.$Q_{cyc}$:Matrix grouping the generalized inertial forces related to Coriolis and Centrifugal acceleration terms.${\vec{Q}}_{grav}$:Vector of generalized gravitational forces.${\vec{Q}}_{act}$:Vector of generalized active forces.$J_{q}$:Constraint Jacobian matrix.$\vec{\lambda }$:Vector of Lagrange multipliers.

In order to avoid $\vec{\lambda }$, which are unknown variables, an orthogonal complement $R^{*}$ is considered [33], verifying that:(4)$$R^{{*}^{T}}\cdot J_{q}^{T}=\vec{0}$$

The Jacobian matrix $J_{q}$ expresses the derivatives of the geometrical constraints on both the independent and secondary variables. The first columns refer to the M=11 secondary generalized coordinates, and the last columns correspond to the F=4 independent variables. Equation (Equation 4) is fulfilled by defining $R^{{*}^{T}}$ as:(5)$$R^{{*}^{T}}={\left[\begin{array}{c} -{\left[J_{q}^{s}\right]}_{M\times M}^{-1}\cdot {\left[J_{q}^{ind}\right]}_{M\times F}\\ 1_{F\times F} \end{array}\right]}_{N\times F}$$

Multiplying both sides of the expression (Equation 3) by $R^{{*}^{T}}$, the dynamic model could be defined in terms of the four independent generalized coordinates (F=4), leading to:(6)$${\left(R^{*}\right)}_{F\times N}^{T}\cdot \left(M_{N\times N}\cdot {\vec{\ddot{q}}}_{N\times 1}+Q_{cyc_{N\times N}}\cdot {\vec{\dot{q}}}_{N\times 1}+{\vec{Q}}_{grav_{N\times 1}}\right)={\vec{\tau }}_{F\times 1}$$
where $\vec{\tau }$ is the vector of forces applied by the actuated joints. Due to the architecture of the 3UPS+RPU PR, the most important friction forces are located at the screw-ball prismatic actuators (${\vec{F}}_{f}$) and can be added to the reduced dynamic model as follows:(7)$${\left(R^{*}\right)}_{F\times N}^{T}\cdot \left(M_{N\times N}\cdot {\vec{\ddot{q}}}_{N\times 1}+Q_{cyc_{N\times N}}\cdot {\vec{\dot{q}}}_{N\times 1}+{\vec{Q}}_{grav_{N\times 1}}\right)+{\vec{F}}_{f_{F\times 1}}={\vec{\tau }}_{F\times 1}$$

Finally, grouping each term using ${\left(R^{*}\right)}_{F\times N}^{T}$ in the reduced dynamic model, Equation (Equation 7) is rewritten as follows:(8)$$M\left(\vec{q}\right)\cdot \vec{\ddot{q}}+C\left(\vec{q},\vec{\dot{q}}\right)\cdot \vec{\dot{q}}+\vec{G}\left(\vec{q}\right)+{\vec{F}}_{f}\left(\vec{\dot{q}}\right)=\vec{\tau }$$

According to [34], the dynamic model of the 3UPS+RPU PR considering a linear friction model can be written linearly with respect to a set of parameters, that is:(9)$$M\left(\vec{q}\right)\cdot \vec{\ddot{q}}+C\left(\vec{q},\vec{\dot{q}}\right)\cdot \vec{\dot{q}}+\vec{G}\left(\vec{q}\right)=K_{rb}\left(\vec{q},\vec{\dot{q}},\vec{\ddot{q}}\right)\cdot \vec{\Phi_{rb}}$$
(10)$${\vec{F}}_{f}\left(\vec{\dot{q}}\right)=K_{f}\left(\vec{\dot{q}}\right)\cdot {\vec{\Phi }}_{f}$$

Applying Equations (Equation 9) and (Equation 10) to (Equation 8), it becomes:(11)$$\vec{\tau }=\left[\begin{array}{cc} K_{rb}\left(\vec{q},\vec{\dot{q}},\vec{\ddot{q}}\right) & K_{f}\left(\vec{\dot{q}}\right) \end{array}\right]\cdot \left[\begin{array}{c} {\vec{\Phi }}_{rb}\\ {\vec{\Phi }}_{f} \end{array}\right]$$
where ${\vec{\Phi }}_{rb}$ and ${\vec{\Phi }}_{f}$ are the rigid body dynamics and friction parameters, respectively. $K_{rb}$ and $K_{f}$ are a matrix related to the dynamic model and linear friction model, respectively. Regarding the friction model, the Coulomb and viscous linear model [35] is considered. The friction model was selected based on the requirements of velocity for knee rehabilitation and diagnosis tasks.

### 2.4. Gravitational Component

Knee rehabilitation and diagnosis is performed at a low velocity, e.g., choosing 20 mm · s${}^{-1}$ for translational movements and $2^{\circ }\cdot$ s${}^{-1}$ for rotational motion. Figure 3 shows the control actions $\vec{\tau }$ for the PR’s limb 2 in a knee flexion–extension combined with hip flexion and knee rotation [36]. Figure 3a verifies that the dynamic behavior of the PR is mainly affected by the gravitational component $\vec{G}$ and the friction effect on the actuated joints ${\vec{F}}_{f}$. In addition, Figure 3b shows a low contribution of the inertial terms (${\vec{F}}_{in}=M\left(\vec{q}\right)\cdot \vec{\ddot{q}}$) and the Coriolis and centrifugal terms (${\vec{F}}_{cyc}=C\left(\vec{q},\vec{\dot{q}}\right)\cdot \vec{\dot{q}}$).

For knee rehabilitation tasks, the term $\vec{G}$ changes according to the weight of the patient’s leg; thus, this research focuses on the identification of gravitational parameters. That is:(12)$$\vec{G}{\left(\vec{q}\right)}_{F\times 1}=K_{G}{\left(\vec{q}\right)}_{F\times 36}\cdot {\vec{\Phi }}_{G_{36\times 1}}$$
(13)$$\begin{array}{c} {\vec{\Phi }}_{G}=[m_{11}\;m_{11}x_{G_{11}}\;m_{11}y_{G_{11}}\;m_{11}z_{G_{11}}\;m_{12}\;m_{12}x_{G_{12}}\;\;\;\;\;\;\;\;\;\;\\ m_{12}y_{G_{12}}\;m_{12}z_{G_{12}}\cdots m_{m}\;m_{m}x_{G_{m}}\;m_{m}y_{G_{m}}\;m_{m}z_{G_{m}}{]}^{T} \end{array}$$
where $m_{ij}$, $m_{ij}x_{G_{ij}}$, $m_{ij}y_{G_{ij}}$, $m_{ij}z_{G_{ij}}$ are the masses and the first moments of mass in components *x*, *y*, and *z*, respectively. The subindex *i* indicates the limb of the PR, and the subindex *j* represents the cylinder (j=1) or rod (j=2) of the limb. The subindex *m* is related to the mobile platform properties. The first moments of mass are measured with regard to a local reference system so that they do not match the centroid of each link.

Finally, considering the terms $\vec{G}$ and ${\vec{F}}_{f}$ as the main dynamic components, Equation (Equation 11) becomes:(14)$$\vec{\tau }=\left[\begin{array}{cc} K_{G}\left(\vec{q}\right) & K_{f}\left(\vec{\dot{q}}\right) \end{array}\right]\cdot \left[\begin{array}{c} {\vec{\Phi }}_{G}\\ {\vec{\Phi }}_{f} \end{array}\right]$$

### 2.5. Base Parameter Identification

Dynamic parameter identification may be performed by two methods [37]:Direct: Performing a single experiment to identify all the parameters of the dynamic model simultaneously. For the PR under study, this method involves identifying ${\vec{\Phi }}_{G}$ and ${\vec{\Phi }}_{f}$ simultaneously.Indirect: Developing different experiments to sequentially identify the parameters, e.g., first the ${\vec{\Phi }}_{f}$ is identified and then the ${\vec{\Phi }}_{G}$ is identified.

The friction parameter identification depends on factors such as the joint surface condition, joint temperature, and lubricant distribution. According to [34], in PRs with a prismatic actuator, the friction effect on actuators makes it difficult to identify the rigid body parameters. Based on this assumption, in this study, we selected the indirect identification of ${\vec{\Phi }}_{G}$, where a linear friction model is previously identified.

The experiment used to define the friction model was carried out independently for each actuator and is based on the fact that the friction term can be approximated by subtracting the gravitational force to the total force using (Equation 8) and ignoring the inertial and centrifugal and Coriolis terms, as justified in Figure 3. This strategy was followed because measuring directly the friction force is far more complex than sensing the gravitational component, and also the total force is directly provided by the control computer.

To this end, a force sensor was attached on top of the actuator (replacing the spherical joint), corresponding to a 0–100 kg load cell of model JLBM, capable of measuring the force on its axis, which was vertically aligned together with the actuator. This setup allows to perform experiments with different payloads attached to the sensor. The gravitational term comes from the readings of the sensor, which accounts for the external payload applied to the actuator and the acceleration due to the trajectory. However, to this term must be added the force of the rest of elements displaced by the actuators, corresponding to the weights of the sensor itself and the rod. The experiments were performed in two ways (Figure 4):Compression experiment, where the mass is directly attached to the axis of the sensor (Figure 4a).Traction experiment, where a system of pulleys is used to provoke traction force with the payload attached (Figure 4b).

**Figure 4 sensors-23-02790-f004:**
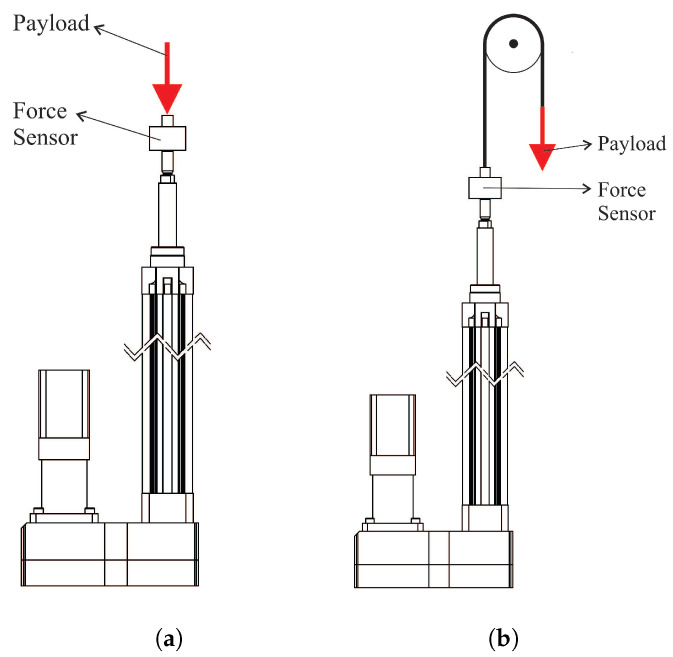
Friction experiments for (**a**) compression and (**b**) traction.

This experiment allows to determine the effects of different masses on the friction response as depicted in Figure 5, where several masses between 1 and 4 kg were used for the external actuators, and from 1 to 10 kg were attached to the central actuator, both for compression and traction. This justifies that the friction force depends on the velocity, but not on the magnitude of the mass placed. For the external limbs, the results are similar since the actuator is the same for all of them, while the central limb shares the shape of the curve but with different magnitude. The data have been fitted with a Coulomb and viscous linear model since it is the most widely used model due to its simplicity and it captures the behavior of the signals (see Figure 5).

Usually, the number of parameters to identify is greater than the number of available dynamic equations, making (Equation 14) an undetermined system. By applying (Equation 14) to $N_{pts}$ different configurations of the 3UPS+RPU PR, the following overdetermined system is obtained:(15)$$\vec{T}=W_{G}\left(\vec{q}\right)\cdot {\vec{\Phi }}_{G}$$
with:(16)$$\vec{T}=\left[\begin{array}{c} {\vec{\tau }}_{1}-{\vec{F}}_{f_{1}}\\ {\vec{\tau }}_{2}-{\vec{F}}_{f_{2}}\\ \vdots \\ {\vec{\tau }}_{N_{pts}}-{\vec{F}}_{f_{N_{pts}}} \end{array}\right]$$
(17)$$W_{G}\left(\vec{q}\right)=\left[\begin{array}{c} K_{G_{1}}\\ K_{G_{2}}\\ \vdots \\ K_{G_{N_{pts}}} \end{array}\right]$$
where $\vec{T}$ and $W_{G}$ are observations from linear model (Equation 14) in different configurations.

After applying the SVD procedure to (Equation 15), the parameters ${\vec{\Phi }}_{G}$ are linearly combined to obtain a set of base parameters ${\vec{\Phi }}_{base}$ that simplifies the overdetermined system as follows:(18)$$\vec{T}=W_{base_{(F\cdot N_{pts})\times r}}\left(\vec{q}\right)\cdot {\vec{\Phi }}_{base_{r\times 1}}$$
where *r* is the rank of the observation matrix $W_{G}$. For an arbitrary configuration, the identified ${\vec{\Phi }}_{base}$ determine $\vec{\tau }$ as:(19)$$\vec{\tau }={\vec{F}}_{f}+K_{base}\left(\vec{q}\right)\cdot {\vec{\Phi }}_{base}$$
where $K_{base}$ is defined by the first *r* columns of the product $K_{G}\cdot P$, with *P* as the permutation matrix of the SVD procedure [21]. A detailed explanation of this base parameter identification procedure is presented in [34].

### 2.6. Offline Identification Results

The workspace of a PR is smaller than that of a serial one, and a PR usually has Type II singularities inside its workspace, reducing its capacity to develop exciting trajectories. In consequence, the first base parameter identification performed in this study considers nine fundamental knee rehabilitation trajectories [36] for the 3UPS+RPU PR (offline identification). In this offline identification process the vector of forces $\vec{T}$ is obtained by stacking together several samples coming from the subtraction of the friction force to the total control action in $N_{pts}$ time steps, according to (Equation 16). Note that the friction force is calculated with the Coulomb and viscous model as defined in Section 2.5 using the velocity provided by the control computer. On the other hand, the observation matrix $W_{G}$ also collects data from those time steps and depends on the generalized coordinates ($\vec{q}$) as expressed in (Equation 17), which are obtained from the active generalized coordinates (${\vec{q}}_{ind}$) measured by the encoders.

The trajectories define a non-singular $W_{G}$ matrix, which has 6 null columns related to 6 parameters of ${\vec{\Phi }}_{G}$. These 6 parameters, with no dynamic effect, are not included in the $W_{G}$ matrix; thus, the SVD identification procedure starts with n=30 parameters. After applying the SVD procedure to the 3UPS+RPU PR, r=15 base parameters are identified, see Table 2.

In addition, we applied the inertia transfer procedure [38] to the PR under study. In this way, the symbolic equations of each base parameter are obtained. Table 3 shows the results for base parameters 9 and 10. The inertia transfer method verifies the physical sense of the base parameter identified by the SVD procedure.

Despite the original matrix $W_{G}$ being non-singular, it has a condition number of $8.64\times 10^{33}$, which is extremely high for effective identification with experimental data. After estimating the 15 base parameters, the condition number of the matrix $W_{base}$ is reduced to 647, which is still high. The architecture of the 3UPS+RPU PR has been optimized [39], and even so, its workspace is not free from Type II singularities. Therefore, the optimization of the rehabilitation trajectories to reduce the $cond[W_{base}]$ is not a practical option. Here, the reduction of the $cond[W_{base}]$ is achieved by eliminating the base parameters with less effect on the dynamic behavior of the PR under study.

The reduction is performed by estimating the accuracy (in percentage) gained in the estimation of the gravitational term $\vec{G}$ when using a subset of $r^{\prime }<r$ base parameters. The resulting gravitational term is $\vec{G}{}^{\prime }=K_{G_{F\times r^{\prime }}}^{\prime }\cdot \vec{\Phi }{}_{G_{r^{\prime }\times 1}}^{\prime }$, and the influence of the $r^{\prime }$ selected parameters is calculated as:(20)$$\mathrm{influence}(\%)=\frac{\sqrt{\vec{G}{}^{\prime }{}^{T}\cdot \vec{G}{}^{\prime }}}{\sqrt{\vec{G}{}^{T}\cdot \vec{G}}}\cdot 100$$

This expression can be applied by starting from $r^{\prime }=1$ and gradually increasing $r^{\prime }$ to see how much influence is gained after the addition of each base parameter. For a value of $r^{\prime }=8$, the gravitational forces $\vec{G}$ can be estimated with 95% accuracy, and Figure 6 breaks down the importance of each parameter in the estimation of $\vec{G}$ in descending order. With this setup, the condition number gets down to 74. This set of 8 base parameters is considered in this study as the relevant parameters (${\vec{\Phi }}_{rel}$).

The identification results for the eight relevant parameters are presented in Table 4. The simulated and actual results are presented in the second and third columns, respectively. All the values in the Table are expressed in SI units.

For an arbitrary configuration of the PR, the term $\vec{G}$ is estimated using ${\vec{\Phi }}_{rel}$ as follows:(21)$$\vec{G}{\left(\vec{q}\right)}_{F\times 1}\approx K_{rel}{\left(\vec{q}\right)}_{F\times 8}\cdot {\vec{\Phi }}_{rel_{8\times 1}}$$
where $K_{rel}$ is a matrix comprising all the rows and columns of $K_{base}\left(\vec{q}\right)$ associated with ${\vec{\Phi }}_{rel}$.

## 3. Model-Based Control

In this Section, we describe the control of the 3UPS+RPU PR with online identification of the relevant parameters. These parameters have already been appropriately expressed so that they can be estimated accurately by using (Equation 21). However, when a human limb is placed on the PR, only the first relevant parameter changes because of the modification of the mobile platform’s mass (row 1 of Table 4). Therefore, the matrix $K_{rel_{F\times 8}}$ should be partitioned into two submatrices $K_{E_{F\times EP}}$ and $K_{NE_{F\times NEP}}$ (which will be denoted simply as $K_{E}$ and $K_{NE}$ unless size matters), where EP denotes the number of estimated parameters, and NEP=8−EP is the number of non-identified parameters, as they are supposed to be known and invariant. Similarly, the vector of parameters ${\vec{\Phi }}_{rel_{8\times 1}}$ is divided into ${\vec{\theta }}_{E_{EP\times 1}}$ and ${\vec{\theta }}_{NE_{NEP\times 1}}$. These are the true values, unknown for the case of ${\vec{\theta }}_{E}$; thus, the controller estimation of this vector is referred as $\vec{\hat{\theta }}$. Note that as the number of identified parameters increases, some estimation precision is lost if these are invariant because unnecessary uncertainty is being added. Nevertheless, the experiments will show how the controller reacts when more than one parameter varies. The experiments below show the robustness of the methods presented in this paper.

Section 3.1 briefly presents one of the current control systems used to estimate parameters online, which is applied to the 3UPS+RPU PR with a reduced set of relevant parameters. Based on this idea, Section 3.2 and Section 3.3 propose two new models of control schemes where relevant parameter identification is performed online directly by regression and least squares estimation techniques, where the estimator parameters are straightforward to tune.

### 3.1. Adaptive Controller

As a preliminary step, let us consider a well-known adaptive controller proposed by Bayard & Wen [40], which has been successfully applied to PRs, e.g., in [25]. The adaptation law is applied along with an underlying PD controller and can be written as follows:(22)$${\vec{\tau }}_{c}=Y\left(\vec{q}\right)\cdot \vec{\hat{\theta }}\left(t\right)+{\vec{G}}_{NE}\left(\vec{q}\right)-K_{d}{\vec{\dot{q}}}_{ind}-K_{p}\vec{e}$$
(23)$$\frac{d\vec{\hat{\theta }}\left(t\right)}{dt}=-\Gamma_{0}\cdot Y^{T}\left(\vec{q}\right)\cdot {\vec{s}}_{1}$$
(24)$${\vec{s}}_{1}=\vec{\dot{e}}+\lambda_{1}\cdot I\cdot \vec{e}$$
with
*Y*:Regressor matrix that participates in the estimation of ${\vec{\theta }}_{E}$. This matrix is implicitly multiplied by $R^{{*}^{T}}$, and is calculated as $Y=K_{E}$.${\vec{G}}_{NE}$:Non-adaptive gravitational term, also comprising the effect of $R^{{*}^{T}}$, and it compensates for the unidentified, known relevant parameters: ${\vec{G}}_{NE}=K_{NE}\cdot {\vec{\theta }}_{NE}$.$K_{p},K_{d}$:Proportional and derivative gains of the PD controller.$\Gamma_{0}$ and $\lambda_{1}$:Matrix and a scalar, which define the dynamics of the estimation process, acting like observer parameters.

The initial estimation of ${\vec{\theta }}_{E}$ is set to the values obtained by the first EP rows of Table 2. The next rows, up to the eighth, are used to define the constant vector ${\vec{\theta }}_{NE}$. The last seven rows are not used because they only minimally affect the dynamic behavior of the PR under study. It is worth mentioning that the parameters $\Gamma_{0}$ and $\lambda_{1}$ are not intuitive to adjust and it is based on trial and error.

### 3.2. Online Identifier with Window-Based Least Squares (WLS)

This method performs a least squares estimation of the relevant parameters when certain conditions are satisfied. First of all, the control law of Equation (Equation 22) remains the same, i.e., there is a PD controller where gravity is compensated via two terms: an adaptive one given by the regressor matrix *Y* and a fixed one expressed by the vector ${\vec{G}}_{NE}$, which are also calculated similarly. However, the estimation of ${\vec{\theta }}_{E}$ relies on an operation of the form:(25)$${\vec{\hat{\theta }}}_{EP\times 1}=W_{E_{(S\cdot F)\times EP}}^{+}\cdot {\vec{T}}_{E_{(S\cdot F)\times 1}}$$
where the symbol (${}^{+}$) refers to the Moore-Penrose pseudoinverse. *S* is the first predefined parameter indicating the size of the window or the number of samples to collect before taking the pseudoinverse (which will be stacked, providing that they improve the information given by the followed trajectory, as will be discussed shortly), and F=4 is the number of actuators, as mentioned in Section 2.3. $W_{E}$ is a matrix generated by concatenating S matrices $K_{E}$ (each of size F×EP) as new valid data from the trajectory is fed to the system:(26)$$W_{E}=\left[\begin{array}{c} K_{E1}\\ K_{E2}\\ \vdots \\ K_{ES} \end{array}\right]$$
where the subscripts 1,2,…,S denote different time steps, which need not be temporally equidistant. Similarly, ${\vec{T}}_{E}$ is a vector comprising the exerted forces discounting the fixed gravity contribution and the friction effect, whose samples temporally match the ones chosen for $W_{E}$:(27)$${\vec{T}}_{E_{(S\cdot F)\times 1}}=\left[\begin{array}{c} {\vec{\tau }}_{E1}\\ {\vec{\tau }}_{E2}\\ \vdots \\ {\vec{\tau }}_{ES} \end{array}\right]=\left[\begin{array}{c} {\vec{\tau }}_{c1}-{\vec{G}}_{NE_{1}}-{\vec{F}}_{f1}\\ {\vec{\tau }}_{c2}-{\vec{G}}_{NE_{2}}-{\vec{F}}_{f2}\\ \vdots \\ {\vec{\tau }}_{cS}-{\vec{G}}_{NE_{S}}-{\vec{F}}_{fS} \end{array}\right]=\left[\begin{array}{c} {\vec{\tau }}_{PD1}+KE\cdot {\vec{\hat{\theta }}}_{1}-{\vec{F}}_{f1}\\ {\vec{\tau }}_{PD2}+KE\cdot {\vec{\hat{\theta }}}_{2}-{\vec{F}}_{f2}\\ \vdots \\ {\vec{\tau }}_{PDS}+KE\cdot {\vec{\hat{\theta }}}_{S}-{\vec{F}}_{fS} \end{array}\right]$$

This method does not perform continuous estimation of the parameters ${\vec{\theta }}_{E}$. To justify this statement, let us suppose the matrix $W_{E}$ is always full, that is, currently estimating the right parameters, but a sudden change of any of them is met. At this instant, new incoming data, of a different nature, will be feeding the matrix $W_{E}$, which will have mixed information from both situations, leading to poor performance that will be challenging to overcome because the upcoming data will also be affected by this drift. Moreover, considering reasons of computational efficiency, the online identifier will only be active when an error threshold is surpassed, in terms of joint positions and references (i.e., the same error used by the PD controller). This error threshold is the second tuned parameter.

In addition to the error-based criterion that initiates the identification process, a stop condition must be designed so that the algorithm knows when the new steady state (as far as the estimation is concerned) is reached. This method proposes a convergence-based stop condition, i.e., it compares the current and previous values of $\vec{\hat{\theta }}$ and, if they differ by less than a convergence threshold, it finishes the process of identification. This threshold is the third parameter involved in the design of the controller. Both error-based and convergence-based conditions can be combined in just one condition statement that starts the algorithm.

Another aspect to consider is the way $W_{E}$ and ${\vec{T}}_{E}$ are filled and cleared. The online identification method uses a strategy based on the condition number: append new data to $W_{E}$ and ${\vec{T}}_{E}$ as long as $W_{E}$ remains well-conditioned after the addition. A fourth parameter indicating the threshold for the condition number of $W_{E}$ is required for such purpose.

However, in some situations the previous constraint may not be enough. For example, when estimating just one relevant parameter (which is generally the case) $W_{E}$ is a vector and, thus, the condition number is always one (because it has just one singular value). To ensure that the matrix $W_{E}$ is filled with samples as independently as possible even in this case, the samples are only collected if a velocity threshold (which is the fifth and last parameter) is exceeded. In other words, a trajectory is being described and the robot is not stuck in a fixed position. In addition, the matrices are cleared gradually (unless already empty), discarding the oldest sample when the error and convergence conditions are not fulfilled.

The algorithm performing the process of identification described above for every time step is given by Algorithm 1.
**PARAMETERS****Variable****Description****Default***S*number of samples400
 et
error threshold0.001
 ct
convergence threshold0.001
 nt
condition number threshold400
 qpt
joint velocity threshold0.001**INPUTS****Variable****Description**
 $K_{E}$
matrix associated with the calculation of estimated relevant parameters
 $\vec{e}$
vector of error between position and reference
 ${\vec{\tau }}_{E}$
exerted force discounting the non-estimated gravity and friction term**OUTPUTS****Variable****Description****Initialization**
 ${\vec{\hat{\theta }}}_{k+1}$
vector of new estimated parameters${\vec{\Phi }}_{rel}\left(1:EP\right)$ or $\vec{0}$**PERSISTENT VARIABLES****Variable****Description****Initialization**
 ${\vec{\hat{\theta }}}_{k+1}$
vector of new estimated parameters (also output)${\vec{\Phi }}_{rel}\left(1:EP\right)$ or $\vec{0}$
 ${\vec{\hat{\theta }}}_{k}$
vector of previous estimated parameters${\vec{\Phi }}_{rel}\left(1:EP\right)$ or $\vec{0}$
 $W_{E}$
input matrix to perform regression concatenating $K_{E}$ matricesempty matrix
 ${\vec{T}}_{E}$
output vector to perform regression concatenating ${\vec{\tau }}_{E}$ vectorsempty vector

**Algorithm 1:** Window-based least squares estimator

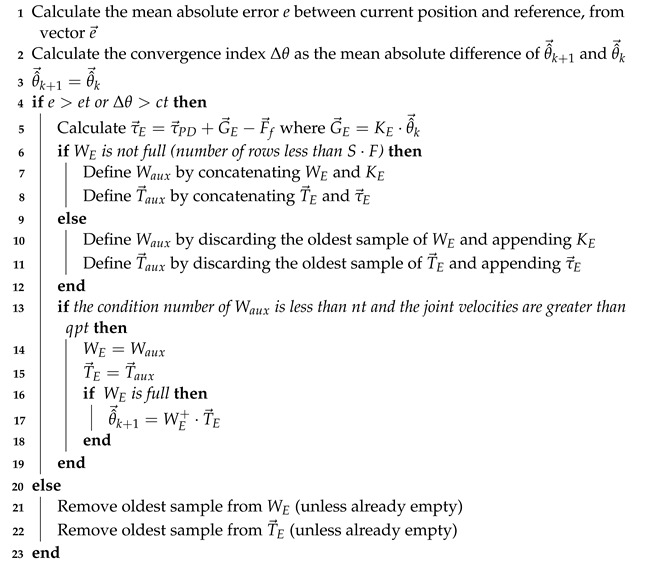



The representation of the controller using this estimator is described in Figure 7, where the vector ${\vec{G}}_{E}=K_{E}\cdot {\vec{\hat{\theta }}}_{k}$ stands for the final gravitational effect due to the estimated relevant parameters in Equation (Equation 22).

The parameter that most affects the proposed control scheme is the number of samples (or window size), as the incoming data ${\vec{\tau }}_{E}$ is usually a noisy signal and robustness is obtained using a large amount of variable data. Thus, there is a tradeoff between the accuracy of the estimation and the time required to update the estimated parameter, since both of them increase with the window size. The thresholds are by default set to 0.001 (in their corresponding SI units) except for the condition number threshold, which is set to 400.

### 3.3. Online Identification Using Recursive Least Squares (RLS)

The recursive least squares method [41,42] computes the new estimate set of the relevant parameters in every iteration using the previous estimate and assuming that they are drawn from a Gaussian distribution. Moreover, the model relating the relevant parameters with the forces ${\vec{\tau }}_{E}$ is linear with white noise $\vec{v}$, whose covariance is a known matrix *V*:(28)$$\vec{\theta }\left(t\right)∼N\left(\vec{\hat{\theta }},\Sigma \right)$$
(29)$$\vec{v}∼N\left(\vec{0},V\right)$$
(30)$${\vec{\tau }}_{E}=K_{E}\cdot {\vec{\theta }}_{E}+\vec{v}$$

By applying Bayesian inference on the distribution of ${\vec{\theta }}_{E}$ assuming constant noise [43], the resulting posterior distribution of ${\vec{\theta }}_{E}$ is also a Gaussian whose estimation of the mean and covariance is given by:(31)$$\Lambda =\lambda_{f}^{-1}\Sigma$$
(32)$${\vec{\hat{\theta }}}_{k+1}={\vec{\hat{\theta }}}_{k}+\Lambda \cdot K_{E}^{T}\cdot {\left(K_{E}\cdot \Lambda \cdot K_{E}^{T}+V\right)}^{-1}\cdot \left({\vec{\tau }}_{E}-K_{E}\cdot {\vec{\hat{\theta }}}_{k}\right)$$
(33)$$\Sigma =\Lambda -\Lambda \cdot K_{E}^{T}\cdot {\left(K_{E}\cdot \Lambda \cdot K_{E}^{T}+V\right)}^{-1}\cdot K_{E}\cdot \Lambda$$

In these expressions, instead of using the value of Σ to estimate ${\vec{\hat{\theta }}}_{k+1}$, a forgetting factor $\lambda_{f}\in \left(0,1\right)$ is introduced to account for the extra uncertainty in the data (because the model is time-variant) and defines a new higher a priori covariance Λ [44]. The initial parameters of the prior distribution are set to ${\vec{\hat{\theta }}}_{0}={\vec{\Phi }}_{rel}\left(1:EP\right)$ or ${\vec{\hat{\theta }}}_{0}=\vec{0}$ and $\Sigma_{0}=10^{-5}\cdot I_{EP\times EP}$, and the noise is considered low non-zero value to avoid matrix singularities: $V=0.01\cdot I_{F\times F}$. The forgetting factor $\lambda_{f}$ usually takes a value between 0.9 and 1 according to the uncertainty and dynamic behavior of the estimation. As $\lambda_{f}$ approaches 1, the estimation gets smoother but slower, as it uses more information from previous data. At the limit of $\lambda_{f}=1$, the estimation becomes inaccurate because no data is forgotten, meaning that it equates a normal regression for which all the past data (before and after the change) is considered. Values close to 1 offer good dynamic behavior of the estimation.

The forgetting factor is the only tunable parameter, which is restricted to a narrow range. Thus, this model is more robust in terms of result variability. No error or convergence checking is required because the update is performed in every iteration. Moreover, an increasing size matrix such as the one used in the least squares window-based method is no longer required.

To account for the uncertainty in the estimation (analogously to the role of the condition number in the window-based method), a variance threshold can be established as a limit for the updated a priori covariance Λ. In particular, the greatest eigenvalue of the matrix shows the highest variance under certain direction, after applying the forgetting factor. If that variance exceeds the threshold, the forgetting factor can be set to 1 to prevent previous useful data from being forgotten.

The pseudocode described in Algorithm 2 is run in every time step, and it implements the recursive least squares identification as follows:
**PARAMETERS****Variable****Description****Default**
 $\lambda_{f}$
forgetting factor0.999
 vt
variance threshold0.001*V*covariance matrix of the noise
 $0.01\cdot I_{F\times F}$
**INPUTS****Variable****Description**
 $K_{E}$
matrix associated with the calculation of estimated relevant parameters
 ${\vec{\tau }}_{E}$
exerted force discounting the non-estimated gravity and friction term**OUTPUTS****Variable****Description****Initialization**
 ${\vec{\hat{\theta }}}_{k+1}$
vector of new estimated parameters${\vec{\Phi }}_{rel}\left(1:EP\right)$ or $\vec{0}$**PERSISTENT VARIABLES****Variable****Description****Initialization**
 ${\vec{\hat{\theta }}}_{k+1}$
vector of new estimated parameters (also output)${\vec{\Phi }}_{rel}\left(1:EP\right)$ or $\vec{0}$
 ${\vec{\hat{\theta }}}_{k}$
vector of previous estimated parameters${\vec{\Phi }}_{rel}\left(1:EP\right)$ or $\vec{0}$
 Σ
covariance matrix of the estimated values
 $10^{-5}\cdot I_{EP\times EP}$


**Algorithm 2:** Recursive least squares estimator

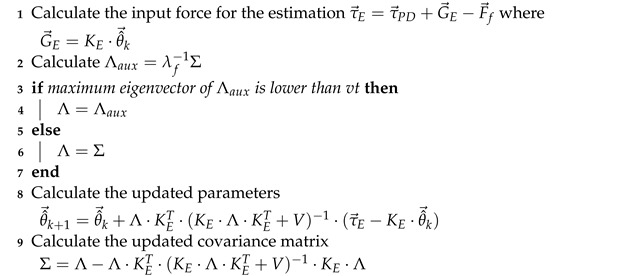



The control scheme is the same as in Figure 7 except for the relevant parameter estimator block.

## 4. Experimental Results

In this Section, we present the results of the relevant parameter identification and control. The fundamental objective is the simultaneous estimation of the gravitational term (i.e., the associated relevant parameters) while keeping the tracking error low thanks to the online adaptation. For that reason, both performance criteria are evaluated and compared. The reliability is also evaluated by measuring the rate of variation of the control action and the rate of variation of the estimation of the relevant parameter. A raw PD+G controller is the baseline for comparing all the methods, which leads to an important consideration: great tracking accuracy can be obtained by just increasing the proportional and derivative gains of the PD. However, this can lead to severe oscillations of the control action, which tries to adapt blindly to any condition, leading to saturations of the actuators and a great sensitivity to noise, stimulating high frequency vibration nodes. This situation becomes inadmissible when it concerns human–robot interaction. This is a justification for why an adaptive control scheme with relatively low gains turns out to be interesting compared with a PD+G. The comparison will be established using the same gains for both controllers: the proportional gains $K_{p}$ are 5800 for the external limbs and 65,000 for the central limb, and the derivative gains $K_{d}$ are 400 for the external limbs and 875 for the central limb. These gains were chosen in a previous experiment with no disturbances in combination with a stationary gravity compensator. Afterwards, the gravity compensator was replaced by the estimator in the experiments with the changing environment.

### 4.1. Experimental Procedure

Each leg of the actual robot is driven by a prismatic actuator, Festo DNCE 32-BS10 for the external limbs ($q_{13}$, $q_{23}$, $q_{33}$) and NIASA M100-F16 for the central limb ($q_{42}$), all attached to Maxon 148867 DC motors, which are commanded by ESCON 50/5 servocontrollers. These controllers offer an accurate current control of the motors, and the current-torque relationship is defined according to their datasheets. DC motors are equipped with incremental encoders with a resolution of 500 counts per turn.

The control unit is connected to an industrial computer through acquisition cards. The reading of the position using the encoder is accomplished through a PCI 1784 Advantech card, with four 32-bit quadruple AB phase encoder counters, while control actions are provided with a 12-bit, 4 channel PCI 1720 Advantech card. The program receives the set of references in the format of generalized active coordinates ${\vec{q}}_{ind_{ref}}$, coming from the solution of the inverse kinematics given the cartesian coordinates of the end-effector. This reference is sampled at a rate of 100 Hz and the positions, velocities, and control actions are stored for posterior analysis.

The controllers described in this paper are implemented in a modular way, using real-time middleware OROCOS (Open Robot COntrol Software) [45] combined with Robot Operating System (ROS) [46], and the C++ programming language.

For these experiments, the trajectories employed involve sinusoidal movements around the generalized coordinates ($x_{m}$, $z_{m}$, θ, ψ), which entail two hip flexions combined with a flexion–extension of knee [36]. Note that different trajectories will stimulate different inertial parameters; thus, mild changes are expected when trying to generalize. Although the trajectories are defined in cartesian coordinates, the sensors and actuators are directly related to joint coordinates, so the first step is to apply Inverse Kinematics to transform the trajectory from cartesian space to joint space. Likewise, the controller works in joint space, so all the performance criteria are also evaluated in joint coordinates. Since the controllers work without a force sensor to estimate the gravitational term, the experiments are performed using payloads of known weight instead of the human limb. This is very convenient to evaluate our controllers because the sensor is not needed and also because it offers much more stable measurements that allow a clearer assessment and comparison of the controllers. However, the same test can be performed with the human limb to perform exercises as explained in Section 2.1. In fact, the trajectory employed is suitable for rehabilitation purposes.

During the experiment, four pauses are introduced where the robot maintains the mobile platform in a fixed horizontal position, waiting for a payload to be placed (or removed) for 20 s before the real rehabilitation exercise is performed. Starting with no payload (stage 1), a 25 kg payload is placed at instant t=100 s (stage 2) and removed at t=220 s (stage 3). Afterwards, a 15 kg payload is placed at t=340 s (stage 4) and removed at t=460 s (stage 5). Figure 8 shows the experiment of mass adition in the real PR.

### 4.2. Model-Based Controllers with Online Identification of the Gravitational Term

In this Section, we compare the results from estimators and error tracking for the two designed controllers and also the baseline, where the gravitational part is calculated by taking the entire set of relevant parameters as fixed.

Regarding the proposed controllers, the rest of the parameters (thresholds) are set to the default values provided in the description of the algorithm. Special attention is required on tuning the window size (set to 400, i.e., the estimation lasts approximately four seconds) and the forgetting factor (0.999) of both controllers, which offer a good tradeoff between the accuracy and timing of the estimations. Most of these constants are initially tuned by a Genetic Algorithm [47] in simulation to establish a fair comparison, and slightly modified in the actual robot.

The trajectory of the performed experiment is shown in Figure 9a for the $q_{23}$ generalized coordinate, during the second stage, in which the 25 kg payload is on the platform. All controllers offer good accuracy, but according to Figure 9b–d, which represent the absolute tracking error of the second generalized coordinate, the adaptive and the proposed controllers outperform the raw baseline controller when the payload is placed on the mobile platform.

From these results, the adaptive and proposed controllers maintain the error below the baseline even when the payload is on the mobile platform. The baseline is more affected because the PD+G considers a constant set of relevant parameters. The results show some peaks of 4.5 mm after a mass addition. The WLS controller also presents some error peaks at the beginning of each stage (see Figure 9b), as it uses this error history to eventually perform a new estimation and improve the tracking a few seconds later.

Table 5 depicts the mean absolute error (MAE) at each stage of the experiment for the second generalized coordinate, and the last row represents the rate of variation of the control action, calculated as follows:(34)$$RV=\frac{1}{n}\sum_{i=2}^{n}\left|\tau_{k}-\tau_{k-1}\right|$$
where *n* is the number of samples of the trajectory, and $\tau_{k}$ is the control action at instant *k*, applied in this analysis by the second actuator. High values of this parameter imply sudden changes of the control action.

All controllers keep the error below 1 mm at stages 1, 3 and 5, in which no payload is used. However, the raw PD+G suffers the effect of the added mass at stages 2 and 4. This effect is compensated for by the rest of the controllers, which keep their errors low. Moreover, the WLS and RLS controllers maintain the nature of the raw PD+G in terms of the smoothness of the control action. The adaptive controller manages to lower the tracking error at the expense of a more fluctuating action according to the last row of Table 5, and this effect is illustrated in Figure 10.

The adaptive controller presents several peaks along the trajectory, while the new controllers remain approximately as smooth as the baseline. This is essentially because both proposed controllers share the fundamental structure of the raw PD+G, while the regression of the adaptive controller is performed via sliding modes. Moreover, the new controllers with online estimator are easier to tune than the adaptive controller, as only one parameter with intuitive meaning (the window size in the case of the WLS controller, or the forgetting factor in the RLS controller) is required in the controller design.

Another important variable to analyze is the estimation of the first relevant parameter, which is illustrated in Figure 11 for the adaptive, WLS, and RLS controllers. The figure shows the effect of both controllers quite well. Starting from a zero-payload reference point of 15.6 kg (according to the first row of Table 4), the payload is changed at 20-s intervals, and then remains constant for 100 s. The window-based controller performs very few estimations; thus, it is suitable for situations where the estimation is intended to be constant until a noticeable change occurs during operation.

However, for dynamic monitoring of the relevant parameter, it would be more interesting to use adaptive or recursive least squares identification, which are capable of updating the relevant parameter immediately, at the expense of a less stable signal.

Rehabilitation exercises can be classified in (i) passive exercises, in which the patient does not exert any effort upon the rehabilitation agent, in this case the mobile platform of the PR. Here, the gravitational term stays more stable and includes the weight of the patient’s leg since no extra force is being applied. In contrast, for (ii) active exercises, the patient must exert extra forces on the mobile platform and the gravitational term is expected to dynamically change during the exercise. The choice of algorithm could be based for instance on this classification. In this regard, a WLS controller would be more suitable for passive exercises, to make very few estimations and only if the gravitational force severely changes, while a RLS controller can perform better in active exercises to catch the dynamic behavior of the changing forces.

Table 6 shows the errors of estimation of the first relevant parameter and the rate of variation with the three controllers (WLS, RLS, and adaptive), and reflects the previous statement: WLS presents a greater error but lower rate of variation, while RLS has a lower error at the expense of an increment in the rate of variation. Overall, RLS outperforms the adaptive controller in the estimation task.

It is interesting to note that all of the controllers approximately converge at the true value when no payload is present. However, the error increases for all of them when the payload is placed on the platform. There are two reasons for this mismatch. Firstly, the standard relevant parameters of Table 4 are estimated using data from a set of various trajectories distinct from the one tested in this experiment; thus, inertial parameters are excited in different ways. Another fact arises when the payload is placed on the platform, as not only is the total mass (represented by the first parameter) increased, but also other inertial properties (centers of mass, moments, etc.) that present a lower contribution to the dynamics are affected, altering, to a certain extent, the value of the other relevant parameters, which are forced to remain constant. The estimation of the first relevant parameter absorbs this effect, which is indeed a good way to avoid reflecting these discrepancies in the tracking error. A recording of the experiment using the RLS controller to estimate the first parameter is available in https://imbio3r.ai2.upv.es/nuevo_video/hybrid-control-identification-one-parameter-high-speed-high-resolution-video-summary (accessed on 22 January 2023).

The last aspect to discuss is the performance of the controllers when trying to estimate a second relevant parameter. In this regard, an additional experiment is conducted using the recursive least squares controller, which is shown in Figure 12. In this experiment, the payload positioning follows the sequence of the previous experiments. However, a second relevant parameter is estimated, which remains constant. The results show that the tracking of the first relevant parameter remains accurate, and the second relevant parameter rapidly agrees with the true value from the beginning of the experiment. The tracking error does not experience any substantial change, and its overall mean absolute value for the second generalized coordinate is 0.80, which is similar to those presented in Table 5. This experiment shows the generalization capabilities of the proposed controllers and can be viewed in https://imbio3r.ai2.upv.es/nuevo_video/hybrid-control-identification-two-parameters-high-speed-high-resolution-video-summary (accessed on 22 January 2023).

## 5. Discussion

This study has proposed two new model-based controllers that estimate certain relevant parameters online involving the PR dynamics. The main gravitational and friction forces have been described, in contrast to the negligible inertial and centrifugal forces for knee rehabilitation and diagnostic tasks. After applying an offline SVD procedure, a subset of 15 gravitational base parameters with a physical sense were identified. These are reduced to a subset of eight relevant parameters by selecting the 95% most dynamic influent parameters. The first relevant parameter includes the mass of the PR’s mobile platform, i.e., the patient’s leg mass can be identified without a force sensor.

The resulting estimation problem becomes straightforward and more robust. Moreover, instructions were embedded into the controller to perform both identification and control tasks simultaneously in rehabilitation tasks. In the literature, some adaptive controllers based on sliding modes have been proposed. However, the proposed approach involves a linear regression, exploiting the linearity property that ties the base parameters and the gravitational forces. The proposed controllers use window-based least squares and recursive least squares, which constitute a new way of performing online identification of a reduced model governed by certain relevant parameters of a PR. To the best of the authors’ knowledge, no previous research has been carried out exploiting the combination of extracting the relevant parameters of a PR to obtain a well-conditioned system and performing a regression of them online in a real experiment in order to (i) keep the tracking error low even during positional changes and (ii) be able to estimate the gravitational force coming from the environment (e.g., the patient’s leg).

This estimation task is ensured to be well conditioned due to the model reduction, and the results demonstrate this by providing outcomes comparable to an adaptive controller with the advantage of easier and more intuitive tuning of its parameters. Moreover, their inherent PD+G structure makes them smoother than the adaptive controller in terms of the control action. The window-based identifier provides a more stable signal at the expense of a slower and intermittent estimation, while the recursive least squares identifier responds dynamically. Moreover, the generalization capabilities were verified when trying to estimate two relevant parameters simultaneously. A ranking of the controllers for each performance criterion evaluated previously is depicted in Table 7, from 1 being the best to 3 being the worst.

In this study, we consider that gravitational terms were expected to change over time; however, they are not the only parameters expected to change over time. Indeed, actuators are prone to deterioration, and friction forces at the joints can change. In future contributions, the proposed identification method will be extended to consider friction parameters, actuator dynamics, or both sets of parameters simultaneously. Since a formal evaluation of the controllers has been provided, the system can also be applied in experiments using a human limb with little or no modification of the algorithms.

## Figures and Tables

**Figure 2 sensors-23-02790-f002:**
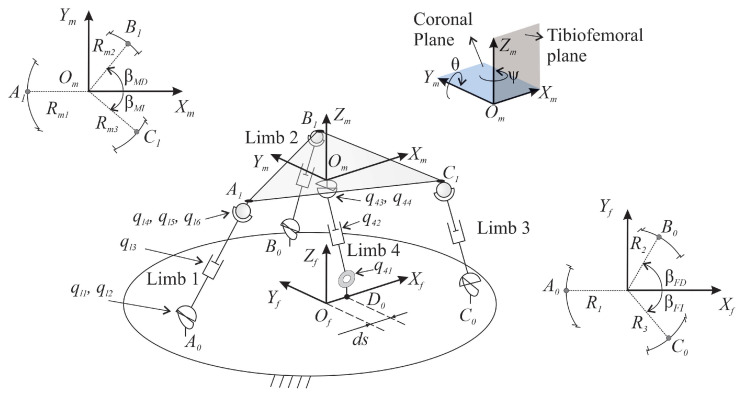
Architecture of the 4-DOF PR.

**Figure 3 sensors-23-02790-f003:**
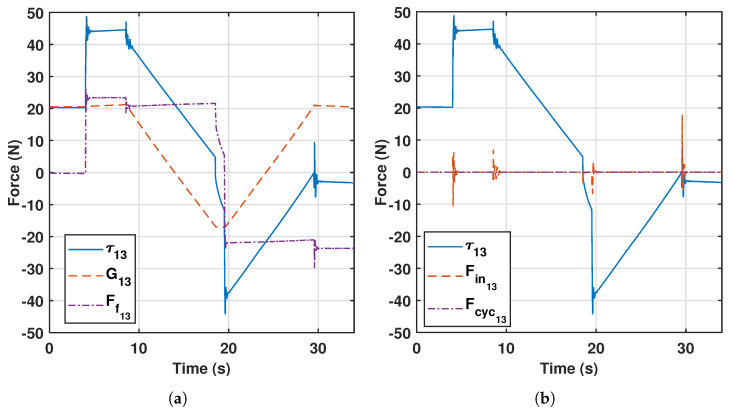
Decomposition of general forces in a rehabilitation trajectory (**a**) $\vec{G}$, $\vec{F_{f}}$; (**b**) ${\vec{F}}_{in}$, ${\vec{F}}_{cyc}$.

**Figure 5 sensors-23-02790-f005:**
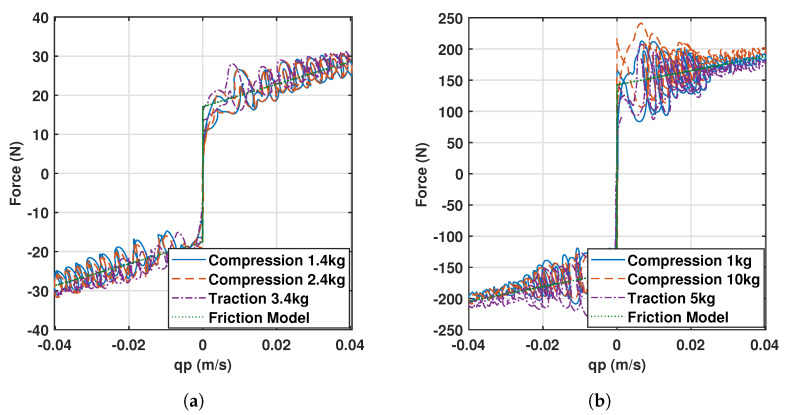
Friction responses and models for (**a**) the external limbs and (**b**) the central limb of the 3UPS+RPU PR.

**Figure 6 sensors-23-02790-f006:**
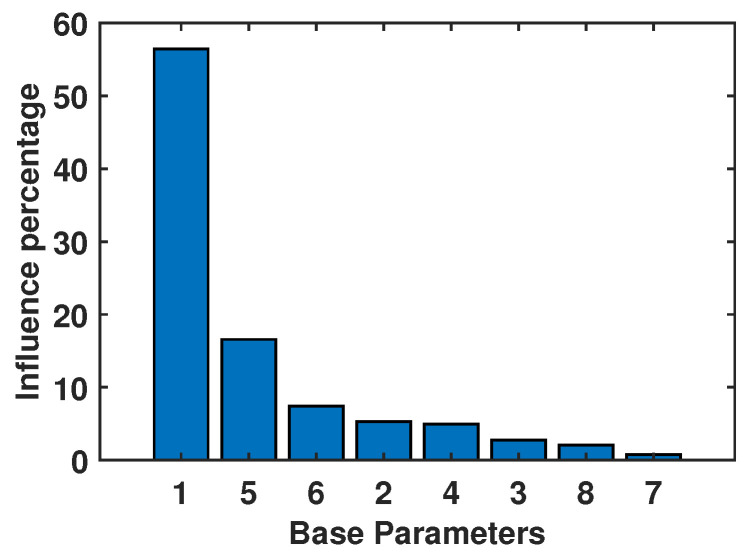
Base parameters with 95% dynamic influence.

**Figure 7 sensors-23-02790-f007:**
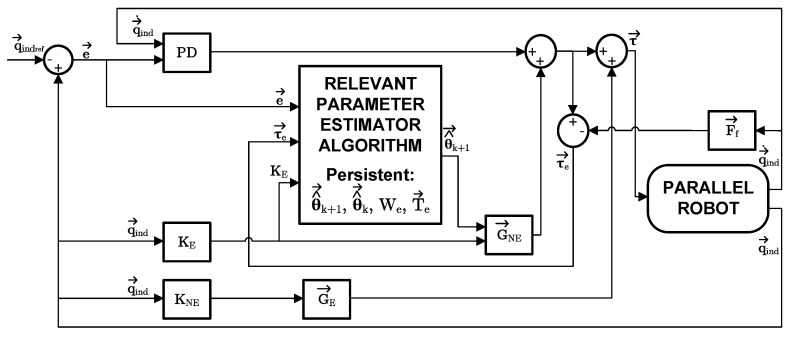
Window-based control scheme with online relevant parameter identifier.

**Figure 8 sensors-23-02790-f008:**
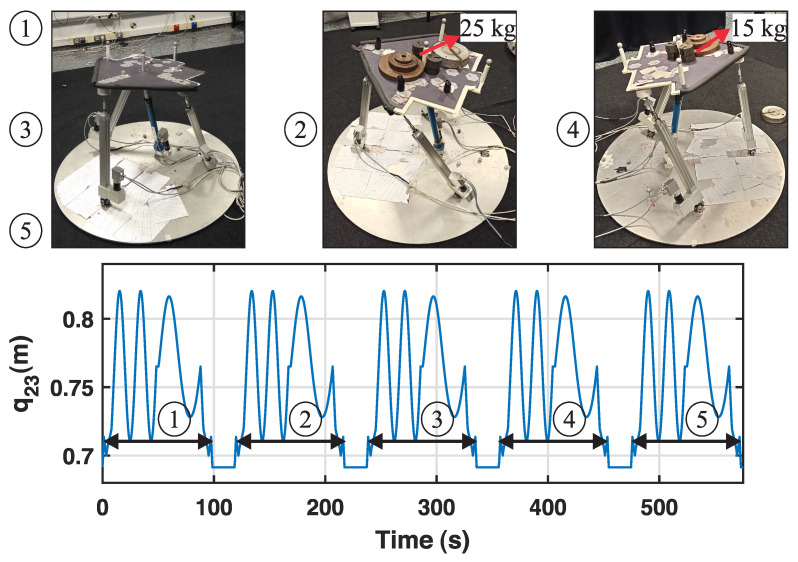
Actual robot and illustration of the stages of the experiment.

**Figure 9 sensors-23-02790-f009:**
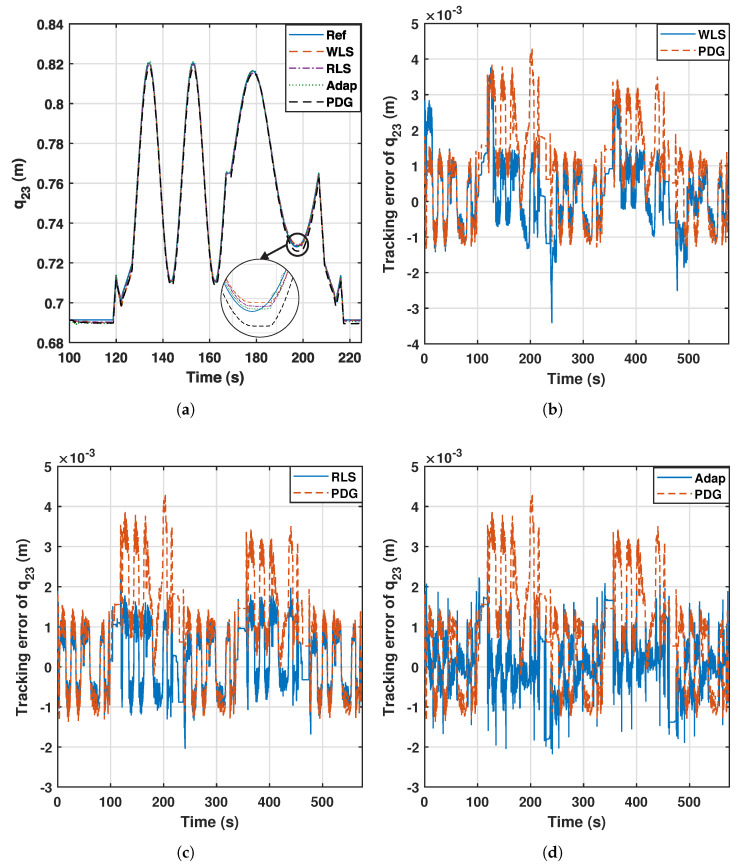
Trajectory tracking of $q_{23}$ and comparison with the PD+G controller.

**Figure 10 sensors-23-02790-f010:**
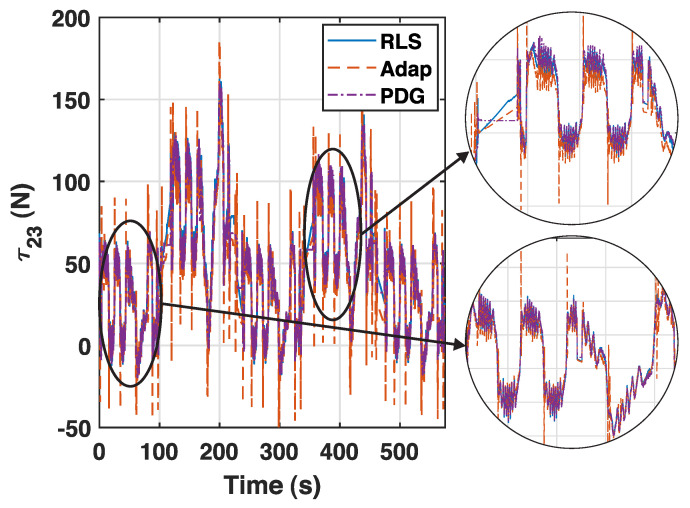
Control actions of the raw PDG, RLS, and adaptive controllers for the second actuator.

**Figure 11 sensors-23-02790-f011:**
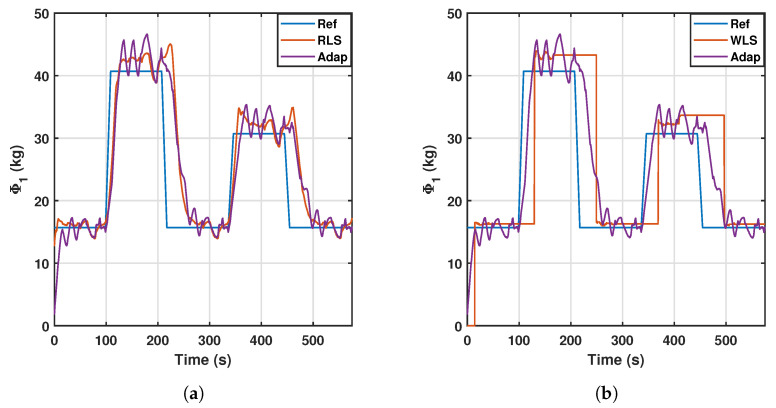
WLS (**a**) and RLS (**b**) estimation of the first relevant parameter. Comparison with the adaptive controller.

**Figure 12 sensors-23-02790-f012:**
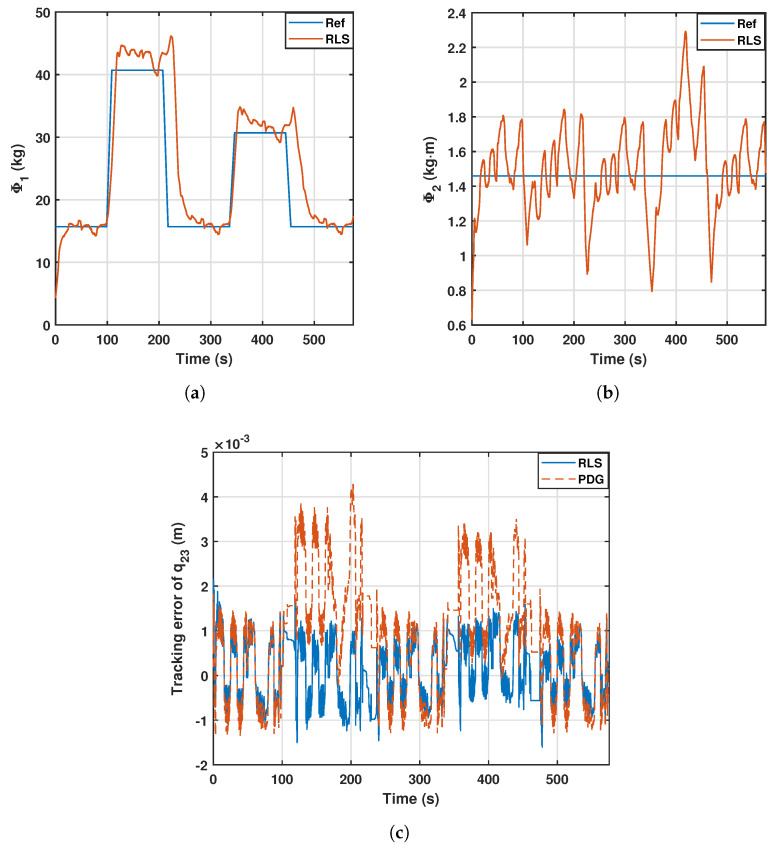
Estimation of the first (**a**) and second (**b**) relevant parameters, and error (**c**) of the second joint with the RLS controller.

**Table 1 sensors-23-02790-t001:** Description of test trajectories for 3UPS+RPU PR.

Movements of Lower Limb	Trajectory of 4-DOF PR
Flexion–extension of knee combined with hip flexion	Elliptical motion on $Z_{f}$ axis as a function of the displacement on the $X_{f}$ axis
Flexion–extension of knee combined with hip flexion, ankle, and knee rotations	Elliptical motion on $Z_{f}$ axis as a function of displacement on the $X_{f}$ axis, simultaneous to rotations θ and ψ
Partial internal–external knee rotation	Rotation ψ with a constant $X_{f}$
Ankle rotation	Rotation θ

**Table 2 sensors-23-02790-t002:** Base parameters for the 3UPS+RPU PR.

No.	Base Parameter
1	$m_{12}+m_{22}+m_{32}+m_{42}+m_{m}$
2	$m_{11}x_{G_{11}}+m_{12}x_{G_{12}}$
3	$m_{21}x_{G_{21}}+m_{22}x_{G_{22}}$
4	$m_{11}z_{G_{11}}+m_{12}z_{G_{12}}$
5	$m_{21}z_{G_{21}}+m_{22}z_{G_{22}}$
6	$m_{31}z_{G_{31}}+m_{32}z_{G_{32}}$
7	$m_{31}x_{G_{31}}+m_{32}x_{G_{32}}$
8	$m_{42}y_{G_{42}}-m_{41}z_{G_{41}}$
9	$0.19m_{22}-0.30m_{12}+m_{m}x_{G_{m}}$
10	$0.23m_{22}-0.30m_{32}+m_{m}y_{G_{m}}$
11	$m_{m}z_{G_{m}}$
12	$m_{42}x_{G_{42}}-m_{41}x_{G_{41}}$
13	$m_{11}y_{G_{11}}+m_{12}y_{G_{12}}$
14	$m_{21}y_{G_{21}}+m_{22}y_{G_{22}}$
15	$m_{31}y_{G_{31}}+m_{32}y_{G_{32}}$

**Table 3 sensors-23-02790-t003:** Symbolic coefficients for the identified base parameters.

No.	Base Parameter	Inertia Transfer
9	$0.19m_{22}-0.30m_{12}+m_{m}x_{G_{m}}$	$R_{m2}cos\left(\beta_{MD}\right)m_{22}-R_{m1}m_{12}+m_{m}x_{G_{m}}$
10	$0.23m_{22}-0.30m_{32}+m_{m}y_{G_{m}}$	$R_{m2}sin\left(\beta_{MD}\right)m_{22}-R_{m3}sin\left(\beta_{MI}\right)m_{32}+m_{m}y_{G_{m}}$

**Table 4 sensors-23-02790-t004:** Relevant parameter identification results for the simulated and actual 3UPS+RPU PR.

No.	Relevant Parameter	Simulation	Actual
1	$m_{12}+m_{22}+m_{32}+m_{42}+m_{m}$	14.12	15.60
2	$m_{11}x_{G_{11}}+m_{12}x_{G_{12}}$	−0.35	1.45
3	$m_{21}x_{G_{21}}+m_{22}x_{G_{22}}$	0.16	0.82
4	$m_{11}z_{G_{11}}+m_{12}z_{G_{12}}$	1.58	0.82
5	$m_{21}z_{G_{21}}+m_{22}z_{G_{22}}$	1.41	−0.89
6	$m_{31}z_{G_{31}}+m_{32}z_{G_{32}}$	1.40	−1.58
7	$m_{31}x_{G_{31}}+m_{32}x_{G_{32}}$	0.15	−0.37
8	$m_{42}y_{G_{42}}+m_{41}z_{G_{41}}$	−0.94	−0.76

**Table 5 sensors-23-02790-t005:** Performance comparison of the tracking of the second generalized coordinate.

Parameter	PDG	WLS	RLS	Adapt.
MAE of tracking of $q_{23}$ in stage 1 (mm)	0.85	0.98	0.81	0.32
MAE of tracking of $q_{23}$ in stage 2 (mm)	2.10	1.19	0.94	0.34
MAE of tracking of $q_{23}$ in stage 3 (mm)	0.85	0.85	0.78	0.32
MAE of tracking of $q_{23}$ in stage 4 (mm)	1.75	1.10	0.90	0.36
MAE of tracking of $q_{23}$ in stage 5 (mm)	0.83	0.86	0.77	0.30
Overall MAE of tracking of $q_{23}$ (mm)	1.27	1.00	0.84	0.33
Overall std error of tracking of $q_{23}$ (mm)	1.32	1.18	0.91	0.45
RV of control action (N)	0.37	0.41	0.39	0.58

**Table 6 sensors-23-02790-t006:** Performance comparison in the estimation of the first relevant parameter.

Parameter	WLS	RLS	Adapt.
MAE of estimation of $\Phi_{1}$ in stage 1 (kg)	0.61	0.65	0.98
MAE of estimation of $\Phi_{1}$ in stage 2 (kg)	6.07	1.96	2.90
MAE of estimation of $\Phi_{1}$ in stage 3 (kg)	0.57	0.83	1.03
MAE of estimation of $\Phi_{1}$ in stage 4 (kg)	4.66	1.71	2.18
MAE of estimation of $\Phi_{1}$ in stage 5 (kg)	0.97	0.78	0.90
Overall MAE of estimation of $\Phi_{1}$ (kg)	3.27	1.59	2.02
RV of estimation of $\Phi_{1}$ (kg $\cdot 10^{-2}$)	0.31	0.38	0.56

**Table 7 sensors-23-02790-t007:** Ranking of performance for each criterion using the analyzed controllers.

Criterion	WLS	RLS	Adapt.
Position tracking	3	2	1
RV of the control action	2	1	3
Estimation of Φ	3	1	2
RV of the estimation of Φ	1	2	3

## Data Availability

Data sharing not applicable.

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
