# Peer review of "Model-Based Control of a 4-DOF Rehabilitation Parallel Robot with Online Identification of the Gravitational Term"

_sensors, 2023, doi:10.3390/s23052790_

Round 1

Reviewer 1 Report

1.  We suggest the authors optimize the size and quality of the figures.

2. The authors must make a better effort at referencing the significant papers. There are several papers published in MDP dealing with knee rehabilitation that are not cited.

3. There are many symbols. A list of symbols (nomenclature) should be given. 

4.  We suggest the authors add a section dedicated to the movement of the knee.

5. We suggest the authors give more details about the restitution of the movements of the knee on this robot.

6.  The knee has 3-DOF, why did you choose a 4-DOF robot?

7.  What are the performance criteria to be met by the proposed controller when using this robot?

8.  The results seem be good, we suggest the authors conduct a comparative analysis and explain the advantages of the proposed controller compared with the existing ones.

9.  The relative works in the following references can be mentioned in the introduction:

-   Brahmia, Allaoua and Kelaiaia, Ridha. "Design of a Human Knee Reeducation Mechanism" Acta Universitatis Sapientiae, Electrical and Mechanical Engineering, vol.11, no.1, 2019, pp.42-53. https://doi.org/10.2478/auseme-2019-0004

- Pisla, D., Nadas, I., Tucan, P., Albert, S., Carbone, G., Antal, T., Banica, A., Gherman, B. Development of a Control System and Functional Validation of a Parallel Robot for Lower Limb Rehabilitation. Actuators 2021, 10, 277. https://doi.org/10.3390/act10100277

-   Brahmia, A., Kelaiaia, R., Chemori, A., and Company, O. (July 5, 2021). "On Robust Mechanical Design of a PAR2 Delta-Like Parallel Kinematic Manipulator." ASME. J. Mechanisms Robotics. February 2022; 14(1): 011001. https://doi.org/10.1115/1.4051360      

Reviewer 2 Report

In this paper, the authors have proposed two new model-based controllers that estimate certain relevant parameters online involving the PR dynamics. The main gravitational and friction forces have been studied. The proposed controllers constitute a new way of performing online identification of a reduced model governed by certain relevant parameters of a PR.

Nevertheless, there are some remarks to the paper.

1. In Figure 5, the authors listed the ranking of base parameters according to their influence percentage on PR dynamic behavior. How this percentage was determined is not explained by the authors. This needs more explanation.

2. In Section 4.2, the authors showed the trajectory tracking effects of each controller in Figure 8a. However, there is no trajectory tracking effect of the WLS controller. This needs to be checked by the authors.

Reviewer 3 Report

In this paper, the authors proposed two model-based controllers with the fundamental PD+G structure applied to a parallel robot for knee rehabilitation. The friction and gravitational parameters are estimated and compensated online. The effectiveness of the proposed controller is verified experimentally and compared with the conventional adaptive controller. However, there are still some problems to be solved:

1. the authors claim that the robot is designed for knee rehabilitation, however, the design and control requirements are not clearly demonstrated based on clinical analysis. Furthermore, no experiments are conducted for knee rehabilitation.

2. what do the X and Y axes represent? Almost all the axis labels of the figures are incomplete.

3. In my opinion, the control object should be the position and the orientation of the mobile platform, however, the experiments only provide the results of a single branch.

4. the parameters of the controllers should be provided.

Round 2

Reviewer 1 Report

Thanks for the point-by-point responses. I don’t have any further comments. One minor suggestion for the title "5. Discussion". I suggest changing it to "5. Discussions and Conclusions."

Reviewer 3 Report

The authors have addressed all my comments, now the paper is acceptable.